# Acceptability of tongue swabs for tuberculosis screening in migrant settings in northern Italy: A qualitative study

Renée Codsi[1☯], Francesca Saluzzo[2,3☯], Rachel C. Wood[1], Alaina M. Olson[1], Giulia Russo[2], Luca Ragazzoni[3], Ramya Kumar[4], George Wanje[5], Marlana Kohn[6], Giovanni Fumagalli[7], Luigi R. Codecasa[7], Gerard A. Cangelosi[1], Daniela Maria Cirillo[2]*

**1** Department of Environmental and Occupational Sciences, School of Public Health, University of Washington, Seattle, Washington, United States of America, **2** Department of Immunology, Transplantation and Infectious Diseases, IRCCS San Raffaele Scientific Institute, Milan, Italy, **3** CRIMEDIM - Center for Research and Training in Disaster Medicine, Humanitarian Aid and Global Health, Università del Piemonte Orientale, Novara, Italy, **4** Department of Epidemiology, School of Public Health, University of Washington, Seattle, Washington, United States of America, **5** Department of Global Health, School of Public Health, University of Washington, Seattle, Washington, United States of America, **6** Department of Health Systems and Population Health, University of Washington, Seattle, Washington, United States of America, **7** Regional T.B. Reference Centre, Villa Marelli Institute/Niguarda Hospital, Milano, Italy

☯ Authors contributed equally to the manuscript
* cirillo.daniela@hsr.it

## Abstract

Human migrations, driven by economic hardship, conflict, and climate change, complicate the global fight against tuberculosis (TB). New strategies are needed to improve the screening of migrants for active TB disease. Current sputum-based testing methods are logistically challenging in many settings. Alternative sampling with tongue swabs is designed to be easier than sputum collection and exhibits acceptable accuracy. This study characterized the acceptability of supervised self-swabbing (SSS) for TB screening in migrant settings in Northern Italy. Migrants arriving through the Central Mediterranean route to Italy were purposely sampled to participate in in-depth interviews (IDIs), which were conducted with the support of a cultural mediator. Data was analyzed using a rapid qualitative analysis approach. The Capability, Opportunity, Motivation-Behavior (COM-B) model guided the systematic assessment of potential barriers and facilitators to SSS. Between November 2023 and June 2024, we conducted 24 IDIs with migrant men and women. Most participants preferred SSS over sputum production and found it relatively easy. Reasons for preferring SSS included its simplicity, privacy, and aversion to sputum collection. Discomfort during swabbing was rare. However, a few participants preferred sputum collection and cited oral hygiene-related complications. Participants highlighted language barriers, trust deficits with the healthcare system, and limited health literacy on infectious diseases, including TB, as factors that could limit the uptake of SSS. Participants also reported that their willingness to participate in TB screening may be driven by a need to comply

**Data availability statement:** Public deposition of these data would breach the conditions of ethical approval and could compromise participant privacy. Researchers who wish to request access to the data for secondary analysis may contact the Segreteria Comitato Etico Territoriale Lombardia 3 (comitatoeticoscientifico@ospedaleniguarda.it) or the University of Washington IRB Administrator, Committee J (hsdteamj@uw.edu). Any data access would require review and approval by the relevant ethics committees. All other relevant data supporting the findings are included within the manuscript and its Supporting Information files.

**Funding:** Copan Italia S.p.A (Brescia, Italy) provided FLOQSwabs® free of charge. Copan had no role in study design, collection, analysis and interpretation of data. This work was supported by the National Institute of Health grant U54EB027049 (RC and GAC) and by the University of Washington Office of the Provost (RC). The funders had no role in the study design, data collection and analysis, discussion to publish, or preparation of the manuscript.

**Competing interests:** The authors have declared that no competing interests exist.

with immigration rules. SSS is a promising and acceptable method for collecting samples for TB screening. To strengthen TB mitigation strategies in this population, future efforts should focus on developing culturally and linguistically tailored educational materials that address the specific concerns and informational needs of migrants.

## Background

Tuberculosis (TB) remains a global health challenge, requiring adaptive strategies for early detection and treatment [1]. Active screening and early case-finding are critical to interrupt transmission chains and prevent disease progression [2]. A major barrier to TB elimination is the underdiagnosis of cases, including those among migrant populations who often face systemic healthcare access challenges. Addressing this issue requires innovative, accessible, and culturally acceptable diagnostic strategies to ensure early detection and treatment initiation [3].

Italy, a low TB-burden country, reported 2,439 TB cases in 2022, 57.4% of which were among foreign-born individuals [4]. Italy, along with Greece, Malta, and Spain, is one of the Mediterranean countries most affected by rising immigration. In recent years, these countries have experienced an increase in both the number and frequency of arriving migrants. According to the United Nations High Commissioner for Refugees (UNHCR) operational data portal, over 43,000 people arrived in Italy via the Mediterranean route in 2024 [5].

Guidelines mandate that healthcare professionals obtain a detailed TB history for all new patients arriving from countries with a high TB-burden (incidence >100/100,000) [6]. Moreover, for people coming from high TB-burden countries, it is recommended to perform screening tests for TB infection, such as the tuberculin skin test (TST) or Interferon Gamma Release Assay (IGRA) [6]. In Italy, migrants coming from high TB incidence countries are obliged to go through TB screening processes. Those who are either symptomatic or chest x-ray positive are asked to provide a sputum sample [6]. When this is needed, the reliance on sputum-based diagnostics poses practical challenges, particularly for individuals who are asymptomatic or unable to produce sputum. This reduces the yield of TB screening [7]. These gaps highlight the need for more adaptable, acceptable, and non-invasive diagnostic approaches to improve TB detection rates among underserved populations.

As potential alternative samples for TB screening, tongue swabs are exhibiting good sensitivity and specificity when paired with a correct collection technique and analytic methods explicitly designed for swab testing [8–16]. A recent study highlighted the ease of collecting TS where 18% of a cohort of mostly asymptomatic household contacts of TB patients were able to produce sputum, while 99% of them were able to provide a TS [17]. The use of TS can facilitate increased diagnostic yields, a measurement where the ability of a participant to provide the requested diagnostic sample is taken into account. Diagnostic yield is important to consider for TB testing given that sputum production is difficult among some populations [18]. Therefore, TS use may offer an advantage in active TB case detection in migrant screening contexts.

Implementation science has shown that the successful implementation of new diagnostic tools requires more than laboratory validation [19]. Persistent gaps exist between the demonstrated efficacy of health interventions in controlled settings and their adoption in real-world public health programs [20]. Integrating tongue swabs into routine TB screening requires acceptability studies to assess migrants' willingness and ability to self-swab, as well as their preferences compared to sputum collection [21]. To date, there is a lack of evidence addressing these aspects among migrant populations, leaving important questions unanswered regarding their preferences, behaviors, and barriers within the context of TB screening.

This study examined user preferences, facilitators, and barriers to tongue swabs for TB screening to support their adoption. We assessed the feasibility and acceptability of supervised self-swabbing (SSS) as a non-sputum alternative for TB screening, identifying key factors influencing its uptake compared to sputum collection in the migrant population.

## Methods

### Participants and setting

This qualitative study was conducted in the Lombardy region of Northern Italy. This is one of the main areas where migrants are relocated upon arrival in Italy. Milan hosts over 50 migrant reception facilities, ranging from large camps to smaller apartments [22]. The screening algorithm includes both active TB and TB infection detection (Fig 1).

### Participant recruitment

Participants were recruited at the Regional Reference Center for TB control in Lombardy, Villa Marelli, Ospedale Niguarda, Milan, where part of the routine screening activities took place. Recruitment started on 24/11/2023 and was completed on 18/06/2024. Individuals who entered Italy through informal routes were the primary target population for the study. To explore migrants' perspectives on TB screening, purposive sampling was used to select participants from these communities who are vulnerable to TB exposure and least likely to seek care due to immigration-related fears, including the perceived risk of deportation. To facilitate outreach and ensure retention, the research team collaborated with migrant welcome centers such as the Red Cross, which provided ongoing support and could assist with follow-up and care for participants who tested positive for TB. Recruitment was conducted with the support of cultural mediators to facilitate participant engagement and ensure inclusiveness. Efforts were made to approach participants from both genders. To promote female participation, female-only migrant reception centers were also contacted.

Inclusion criteria for the interviews included age 18 years or older and having arrived in Italy within the 12 months preceding the interview and residing in the migrant welcome centers. Eligibility was independent of TB screening results; however, participation required willingness to perform self-swabbing and take part in an interview.

The target population primarily included migrants from the Indian Subcontinent (ISC), Southwest Asia and North Africa (SWANA), and West and Central Africa (WCA). These groups have elevated TB risk due to high incidence of TB in their home country as well as migration-related exposure coupled with precarious living and working conditions during the migration journey. Migrants who came through formal migration and who were not living in the welcome centers were excluded, as they faced fewer structural barriers to healthcare and had access to governmental services. Participants were informed about the study objectives and procedures in detail by the interviewers (RC, FS). All potential participants were given sufficient time and opportunity to seek support in their decision-making, including discussing their choice with a trusted person before deciding whether to participate.

### Diagnostic sample collection

Sputum samples were collected through expectoration, according to the available WHO guidelines [23]. Tongue swabs were collected according to the previously published protocol for supervised self-swabbing SSS with the amendment of using the Copan Floqswab with the 80mm breakpoint [9, 24, 25]. The procedure for both the collection of the sputum samples and the tongue swabs is described in the Interview Guide (Supplemental Material I).

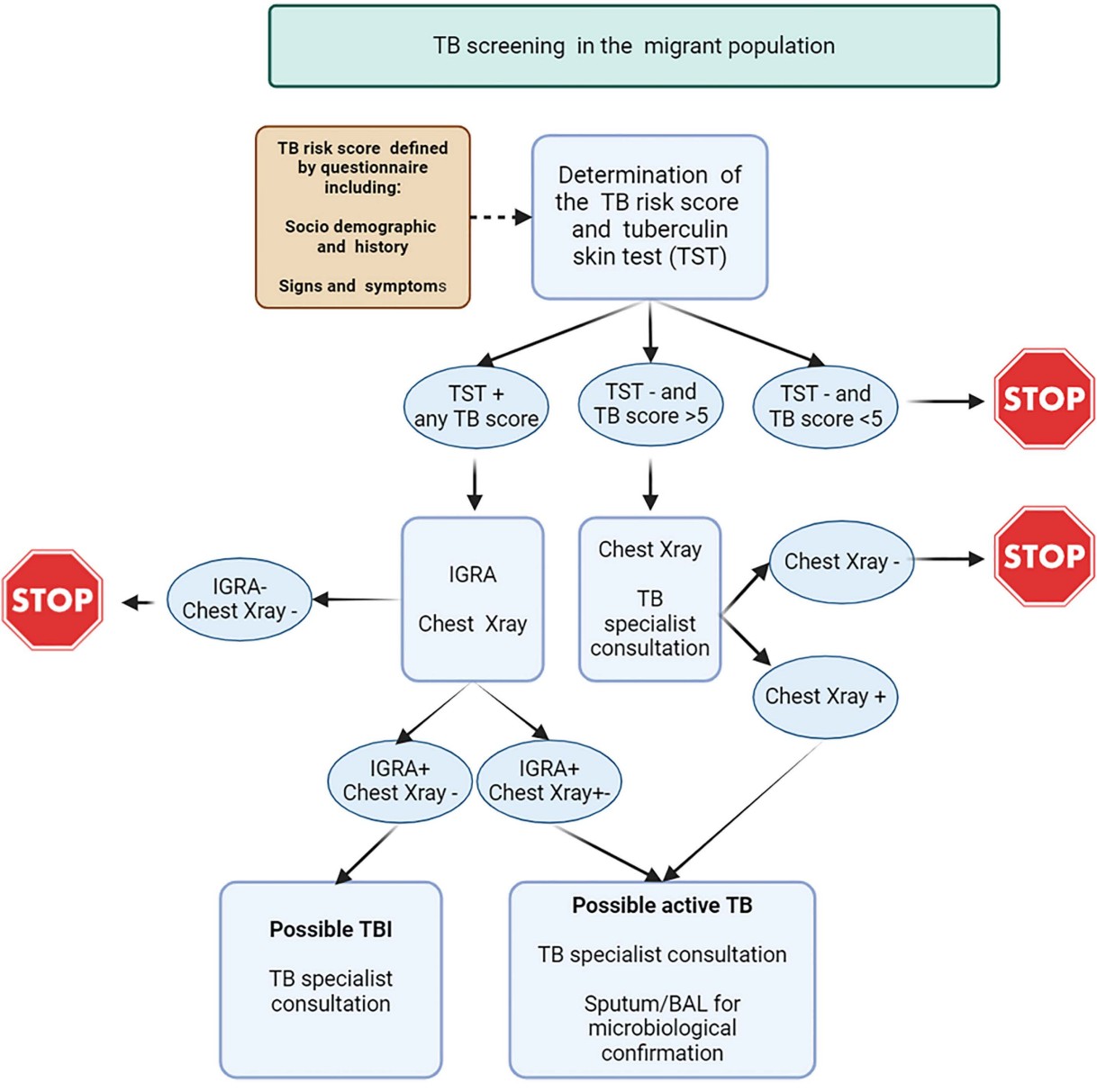

**Fig 1. Lombardy regional TB screening algorithm among migrants coming from high TB incidence countries to Italy.**

## Theoretical model

The study employed the Capability, Motivation, and Opportunity- Behavior (COM-B) model across all study phases, including the design of the study topic guides, data collection, and guiding the analysis and interpretation of the data. The model posits that for a Behavior (B) to take place, an individual must have the Capability (C) to perform the behavior, the Opportunity (O) to engage in it, and the Motivation (M) to initiate and sustain the behavior [26]. Several studies have used the COM-B model to assess the acceptability of health interventions [27–33].

## Data collection

A semi-structured interview guide was developed a priori to explore behavioral determinants influencing the adoption of SSS. The COM-B based interview guide explored participants' capabilities to collect samples according to the SSS protocol. Key considerations included participants' physical ability to use the swab properly without contamination, as well as their comfort level during the procedure. Interview questions also focused on existing or potential opportunities that could enhance the acceptability and willingness to engage in SSS, such as language barriers between migrants and healthcare workers. Barriers and motivations to use SSS were also assessed.

The guide was reviewed by the study team and local investigators to ensure clarity and alignment with study objectives. After deliberations, the refined topic guide was presented to cultural mediators from the cooperative *Farsi Prossimo* in Milan, Italy, which has long-standing experience supporting migrants affected by TB and is the reference cultural mediator cooperative for the regional TB center. We specifically recruited cultural mediators who have a migratory history, come from the same region as the study participants, have experience working in migrant settings, and speak at least one of the participants' languages. We trained them on qualitative data collection methods, and positionality and sensitivity to the migrants' needs. The review by cultural mediators ensured that cultural appropriateness across the diverse migrant groups was met. Interviews were conducted in person by two researchers (RC, FS) at the regional TB reference center. To address language and cultural barriers, trained cultural mediators facilitated real-time translation. Interviews lasted between 60–90 minutes and were audio recorded. Field notes were also taken in real time.

## Data analysis

Translated interviews were analyzed by RC and FS employing a rapid qualitative analysis approach, adapting the deductive framework-based method [34–36]. This method allowed for efficient yet rigorous data processing by using structured notes and a matrix-based approach. The analysis followed a two-step process. First, rapid data extraction and categorization were conducted by each researcher, entering their interview notes into a structured research matrix according to Hamilton's rapid qualitative analysis method. The matrix categorization was primarily based on the discussion guides and the thematic concepts of interest. Each transcript was reviewed by 2 people (RC and FS) and no discrepancies were identified between coders. A second analyst then cross-checked the data by listening to the audio recordings and refining the matrix. A thematic analysis was performed using the COM-B model. Responses were classified under capability, which included knowledge about TB and understanding of self-swabbing procedures; opportunity, which encompassed logistical barriers and structural support; and motivation, which involved perceived risk and trust in the method. To support each identified theme, we embedded direct quotations from interviews, ensuring that participant perspectives were accurately represented and contextualized within the COM-B framework (Fig 2). Interpretation of thematic analyses was discussed among reviewers to obtain a consensus. This approach enabled us to systematically assess how risk perception, self-efficacy, and contextual factors influenced engagement with self-swabbing. The rapid analysis reduced resource demands while maintaining analytical depth, making it particularly suited for evaluating implementation determinants in time-sensitive public health settings [34]. Theme saturation was determined based on established qualitative research standards, where homogenous groups typically reach theme saturation after 9–17 interviews [37–41]. Given the diversity of our sample, we anticipated a higher threshold for thematic saturation. Saturation was defined as the point where no new themes emerged from additional interviews [42]. The research team ensured the sample captured the overall variation in responses relevant to the study objectives. The study followed the Consolidated Criteria for Reporting Qualitative Research (COREQ) guidelines [43].

## Ethical approval

The study was approved by the Comitato Etico Territoriale Lombardia 3 (#3582_S_N), recognizing the TB regional reference center Villa Marelli as the study site and IRCCS Ospedale San Raffaele as the sponsoring institution. Ethical

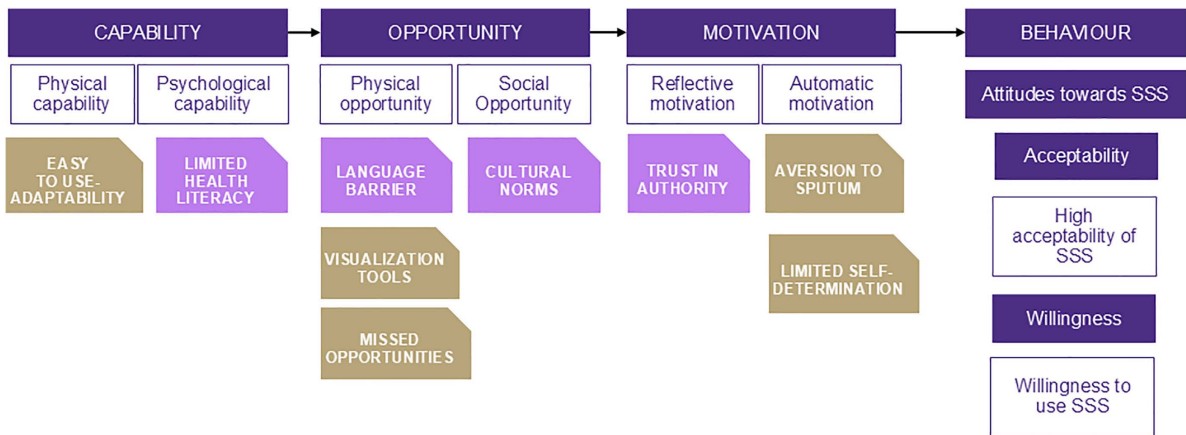

**Fig 2. COM-B analysis of barriers and facilitators to supervised self-swabbing for TB Screening.**

approval was also obtained from the University of Washington Human Subjects Division (UW STUDY00018900). Participants provided both written and oral informed consent. They were explicitly informed of their right to decline any question and withdraw from the study at any time. Procedures for data management and approval for audio recording of interviews were also explained.

## Results

### Participants characteristics

Table 1 describes characteristics of the 24 migrants who participated in the interviews. The sample comprised 19 males and 5 females. Approximately half of the participants were aged between 18 and 30 years. The majority were from the West and Central Africa (WCA) region (54%), followed by the Indian Subcontinent (ISC) region (38%). Regarding marital status, most participants identified as either single or married, and several had children. All participants were asked to provide a sputum sample and to follow the SSS protocol. All participants were able to provide a tongue swab sample; however, only one participant successfully produced a viable sputum sample.

### The capability to use tongue swabs is influenced by their ease of use

Tongue swabs emerged as a practical and preferred alternative to sputum collection for TB screening in migrant settings. Most participants reported that the tongue swab method was simple to perform, and the protocol was easy to follow. Many described the process as "quick," "clean," and "less stressful" than producing sputum.

> "*The (tongue) swab was easier than the sputum because when I tried [producing] sputum I did cough and got some saliva, but I didn't get sputum… but with the (tongue) swab, I was able to do it on my own… it was quick and much cleaner.*" (Male, ISC)

> "*The swab was a lot easier and faster. The sputum process hurts my throat because you have to breathe deeply and cough, and if you can't make the sputum, you have to keep trying… I couldn't cough up anything, so it didn't work.*" (Male, SSA)

Several participants emphasized the value of privacy, autonomy, and simplicity when using SSS for TB screening. Many participants preferred performing the swab themselves to avoid embarrassment or perceived judgment during sputum collection. The convenience of completing the test discreetly and in the same space was a key factor in their preference.

**Table 1. Demographic characteristics of study participants (n = 24).**

| Characteristic | N (%) |
|---|---|
| Age (years) | |
| 18–30 years | 11 (45.8) |
| 31–40 years | 8 (33.3) |
| 41–51 years | 5 (20.8) |
| Gender | |
| Male | 19 (79.2) |
| Female | 5 (20.8) |
| Region of origin | |
| ISC | 9 (37.5) |
| WANA | 2 (8.3) |
| WCA | 13 (54.2) |
| Marital status | |
| Single | 11 (45.8) |
| Married | 10 (41.7) |
| Unknown | 3 (12.5) |
| Children | |
| Yes | 11 (45.8) |
| No | 8 (33.3) |
| Unknown | 5 (20.8) |
| Interview Language | |
| French | 9 (37.5) |
| English | 4 (16.7) |
| Arabic | 2 (8.3) |
| Bengali | 7 (29.2) |
| Urdu | 2 (8.3) |

*"I can do this (SSS) myself. It is easy. I don't want the nurse to do it and be in front of my mouth… I have a problem with really bad breath, I don't know why, it has been since I came here, and I am embarrassed about it. I don't want her (the nurse) to be in front of my mouth when it is something I can do myself. (Male, SSA)*

*"The swab is easier, you do not need to move and go to the other room like you need to for the sputum…and people waiting in line with the cup by the other room, everyone knows you are doing something when they walk by. I don't want them to know what I am doing. I can do it myself here with the (tongue) swab, sitting in front of you…and you can tell me when to start and stop so I know I am doing it right." (Male, SSA)*

*"It is not nice to show your tongue to others. It is not easy for me to open my mouth in front of you. No, not for a physical difficulty in doing it. It is not nice to put something on the tongue in front of others or in any case. I don't want you watching me while I swab." (Male, SSA)*

Participants' previous experiences with self-testing influenced how they perceived the tongue swab method. Some participants felt confident in their ability to self-administer the swab, drawing comparisons to familiar tests like pregnancy kits, which offer immediate results. Few participants had encountered self-testing before this study. However, the lack of visible results with the tongue swab left some feeling disconnected from the outcome. Others questioned the relevance of TB screening, especially if they believed TB was no longer a concern in their country of origin.

*"I have done a self-test for pregnancy before in my country, in the Ivory Coast, and here (in Italy). I did them because I was not feeling well. I did it twice. With the pregnancy test, I can also see the results. I liked that. For the (tongue) swab, I can swab myself fine, but I would like to be able to read the results right away, like I can in the (home) pregnancy test." (Female, SSA)*

*"My mom told me about TB, but since the vaccine campaign for children was started in Bangladesh, we do not see this disease anymore…TB is finished. Old people had it…But it is not a problem in my country, so I don't know why I have to be here for testing. (Male, ISC)*

### Health promotion and guidance on SSS provide an opportunity to increase uptake

Limited health literacy emerged as a key barrier to self-swabbing, reflecting the challenges required for successful implementation. Many participants reported a lack of understanding about TB and the screening procedures, with some expressing confusion and concern about tests they had undergone previously. The lack of detailed information on the purpose and effects of the screening led to anxiety, distrust, and hesitation to engage with healthcare workers. Some participants questioned the safety of the procedures and felt distressed by visible side effects that were not explained to them.

*"On my journey to Italy, I was kidnapped in Libya for ransom… I was around a lot of hurt and sick people who were coughing and moaning in agony all the time, and we were constantly abused… I do not know what TB is or which symptoms TB causes. All I know is that it is an old disease. I know I am here to do the screening to prevent the disease… I am strong, I won't get it (TB) because I have already survived a lot of horrible things". (Male, SWANA)*

*"I don't know what that (TB) is. Two weeks ago, they drew my blood. I have to take medicine because there is a small problem with my lungs. It made me feel bad. I don't know what is going on." (Male, ISC)*

*"I never had TB, and I don't know what it is. They did not explain why I was doing the test (TST) on my arm. Look at the bruise it made (participant points to the wound from the TST test on her forearm)…will it heal and go away soon? It is disgusting, and I am ashamed of it. It makes my arm look ugly and I don't want anyone to see it…What did they give me to make this happen? My arm was fine before they injected things under my skin... When it is hot out and I sweat, it hurts even more." (Female, SSA)*

Participants described language barriers as a major challenge throughout the TB screening process, leading to confusion, fear, and feelings of isolation. Some described feeling like they had no choice but to comply, without fully understanding what was happening. Participants expressed deep relief when finally able to communicate with people in their language.

*"I have so many things to do for my (immigration) papers, I don't know what is happening, and it's all so quick, I just follow the authorities. We are always in a rush when we come with the van and the group from the camp, so we rarely have someone who speaks our language to tell us what is going on and what we need to do, and why and how we can be prepared. Everyone in the van is scared about what they are going to do to us. The rest of them don't speak my languages, so even though there are others with me, I feel alone…" (Male, ISC)*

*"The explanation I received on what TB is and why I am here was not clear. I was scared that she knew I had TB, but I realized only after speaking with you [interviewer] that you are trying to see if I do or not, and that is why I need to try these tests. You're the first person to speak with me in my language. It is such a relief… The nurse tried to tell me what was happening by using Google Translate, but I didn't understand it. Often it translates things in ways that don't mean anything." (Male, ISC)*

Participants highlighted the need for clear, accessible instructions to support SSS. They emphasized that visual tools, like videos, could help overcome literacy and language barriers and improve understanding of the process. Some participants reported that they could not read in any language and would benefit from guided visuals with audio explanations. A few participants noted that while in-person support with translation was ideal, subtitled videos in their native language would also be helpful.

*"If videos were showing me why it's important to do the test and how to swab (SSS) and what to expect after it would be better for me, because even if you can or can't read, you can watch it and do it and understand… I cannot read even in my languages, so I need something helpful with a voice telling me and guiding me with images."* (Male, SSA)

*"If someone explains how I can swab myself for the test by showing a video, it will be better. Ideally, it would be great to have someone like you here to explain to me with a translator, but if you can't be here, then a video in Bengali will also work."* (Male, ISC)

### Motivation to comply with immigration requirements may drive willingness for SSS

Participants described a sense of limited self-determination and uncertainty surrounding TB screening, often linking it to fears about their immigration status. Many participants reported undergoing health checks as part of immigration procedures without fully understanding their purpose, and complied with TB screening because they were told it was mandatory to remain in Italy or access housing and services. This lack of clear communication, combined with language barriers during medical procedures, contributed to confusion and anxiety.

*"They told me the screening was compulsory to stay here in Italy. I don't understand all the things they are doing… They took 3 containers of blood from me and 5 from the man next to me. He was really upset and asked why they took more from him. The nurse didn't understand him, and he was frustrated and walked out. I don't know what is going to happen to him, but I fear for him and hope he doesn't get kicked out. I want to stay in Italy, so I am cooperating."* (Male, SSA)

*"You are the first doctor I speak to with a translator… I come because they take me here from the camp. I don't have a choice."* (Male ISC).

Despite these concerns, participants expressed compliance with administrative requirements to avoid jeopardizing their stay.

*When I arrived here in Italy, the cultural mediator at the camp explained that we had to do the TB screening to be accepted to live at the camp. I came because they told me I had to for my (immigration) papers… I don't want to get in trouble, so I am here."* (Male, ISC)

### Discussion

Our qualitative study explored migrants' willingness to provide tongue swabs compared to producing a sputum sample for TB screening. This study provides novel insights into the acceptability of SSS among migrants in Lombardy, located in Northern Italy. Participants overwhelmingly favored SSS over sputum collection, citing its simplicity, privacy, and ease of use. These preferences reflect both the logistical advantages of SSS and the personal discomfort associated with sputum production, particularly in public or clinical settings. However, the successful implementation of SSS will depend on addressing key barriers identified by participants, including limited health literacy, language challenges, social and cultural

barriers, and mistrust in the healthcare system. Additionally, the motivation to comply with immigration requirements emerged as a powerful driver of participation, highlighting the need for person-centered and contextually responsive strategies to support equitable TB screening in migrant populations.

These findings suggest that flexible, migrant-specific TB screening models are needed.

Guided by the COM-B model, a key facilitator in the capability domain was the ease of use of SSS, described by many participants to be simple and manageable. In the opportunity domain, addressing barriers related to language and lack of information leading to uncertainty about the necessity of TB screening may influence the uptake of SSS. Under the motivation domain, migrants feared repercussions such as deportation, which discouraged them from disclosing health concerns. It also acted as a motivator to comply with administrative requirements for TB screening using SSS. The barriers identified in this study align with broader research on migrants' health-seeking behaviors and barriers to TB screening in Europe [44–49]. A study pooling results from studies in Italy, Sweden, the Netherlands, and the UK found that migrants' trust in healthcare and understanding of TB screening procedures played a crucial role in their willingness to participate in testing [3].

The broader structural inequities in TB care identified in global TB policy discussions are also relevant to this study. A recent paper on equity in TB responses highlights how systemic disadvantages such as poverty, migration status, and social exclusion continue to limit healthcare access for populations most affected by TB [50]. While the End TB Strategy includes an equity target, few efforts have been made to integrate social science and multi-sectoral approaches into TB screening and care. This aligns with our findings that many migrants comply with screening due to a perceived lack of self-determination rather than informed choice. It reinforces the need for equity-driven, community-led interventions to empower migrants in healthcare decision-making. The role of social science in TB research and policy is increasingly recognized as crucial in addressing barriers to care beyond biomedical and public health interventions. A multi-sectoral approach including legal protections, social support policies, and participatory healthcare strategies could help mitigate fear of deportation, increase trust in healthcare systems and providers, and enhance TB screening uptake [51].

Language barriers emerged as a significant challenge to the acceptability and understanding of TB screening among migrants in our study. Participants often reported confusion about the purpose of the screening and described a lack of clear communication during health encounters, which aligns with existing literature highlighting language as a key barrier to accessing care in migrant populations [3]. Similar studies in European and North American contexts have shown that language discordance can lead to reduced health literacy, miscommunication, and mistrust in the healthcare system [3, 52, 53]. Unlike some prior research that focuses on one-on-one HCW support guiding SSS sample collection [24, 25], participants in our study expressed strong preferences for culturally and linguistically tailored videos and visual tools, particularly when interpreter access is limited. Moreover, it was reported that the modality of TB screening delivery impacts uptake, with community-based approaches and culturally tailored engagement strategies yielding higher participation rates [54, 55]. These findings point to the need for innovative, accessible communication strategies in TB screening programs to ensure equitable engagement across diverse language groups.

The introduction of multilingual, culturally adapted health promotion materials, such as instructional videos and illustrations geared at varying literacy levels, could significantly improve participants' willingness and ability to produce a swab sample according to the protocol effectively [24]. Developing SSS-specific health promotion materials tailored to migrants' needs could improve the acceptability of this screening approach. Culturally adapted, multilingual resources should be a priority to increase willingness to seek care, support participation in TB screening, and promote accurate sample collection for diagnosis.

Outside of the study participants' views on TS, the fact that TS have shown lower sensitivities than sputum [8, 12, 56, 57] may hinder its adoption in some settings. However, the use of TS could increase diagnostic yield because TS collection is not incumbent on sputum production. The importance of diagnostic yield has been emphasized recently, particularly in at-risk populations like migrants who may not be able to expectorate sputum [18]. TB screening in a Targeted Universal

TB testing (TUTT) approach, where people are screened based on risk rather than symptom presentation or care-seeking behaviors, would benefit from non-sputum samples like TS [58]. TUTT is an example of a use case where lower sensitivity may be tolerated in favor of potentially higher diagnostic yields [21].

A strength of our qualitative study was the focus on SSS in the migrant population, which has been underexplored. This approach allowed us to capture unique barriers and facilitators specific to this population. By centering the voices of migrants, the study offers practical guidance for designing more inclusive TB screening strategies. In addition, the study was a strong collaboration fostered between developers of tongue swabs at the University of Washington School of Public Health with researchers at the IRCCS Ospedale San Raffaele in Italy. This collaboration enabled both technical expertise and contextual understanding, strengthening the study's relevance and applicability to real-world migrant health settings.

The study also had limitations. First, the recruitment of women was limited as there was a smaller number of female migrants seeking asylum or coming through informal migration and/or residing in the welcome centers. Additionally, many women faced competing demands, such as childcare responsibilities and limited availability of suitable interview settings. The views presented by the women in the study may not be representative of all women, but many of the themes resonate with migrant populations. Despite the underrepresentation of women in the study, our sample size was sufficient to achieve saturation of several themes across interviews. This notwithstanding, our findings may not be generalizable to women in other settings. Secondly, the recruitment of participants from the West Asia and North Africa region and Pakistan was not consistent with the migratory data. It was more challenging to recruit participants from these regions and to provide the appropriate cultural mediators for support. Future studies need to intentionally design recruitment strategies to support hard to reach demographics, such as females and participants from Pakistan and Southwest Asia and North Africa region to understand their perspectives, as they represent a significant portion of the migrant population in Italy who are underserved. This underrepresentation could influence the generalizability of the findings, as the perspectives of women and men from these groups could differ from those represented in the study. Third, social desirability bias may have shaped how participants described their willingness to comfortably provide a sample according to the SSS protocol. To minimize these biases and strengthen trustworthiness, we used trained cultural mediators during interviews, conducted debriefings after each session, and applied systematic credibility checks during analysis. As this was a single-centre study with predominantly male participants, the findings may not capture the full diversity of migrant experiences, particularly those of women, and may not be generalisable to different settings, particularly those that have fewer resources and a higher TB burden. Nonetheless, the themes identified provide useful insights that can inform future research and programme development in European migrant centres.

## Conclusion

Our qualitative findings have shown that adoption of a novel diagnostic SSS is a feasible and acceptable alternative to sputum collection in migrant settings, but its success depends on addressing persistent barriers such as low health literacy, limited trust in healthcare, and cultural acceptability. Tailored, multilingual health promotion tools could improve awareness and confidence in self-swabbing, particularly among migrants. Embedding person-centered, equity-focused strategies in TB screening programs is essential to advancing more inclusive and effective TB control. These findings support the need for the development of public policies and health promotion materials that integrate multilingual, culturally adapted audiovisual materials into TB screening programs to enhance accessibility, engagement, and uptake among diverse migrant populations.

## Supporting information

**S1 Text. Semi-structured interview guide for interviews with migrants: User Acceptance of Migrants Using Tongue Swabs for TB Sample Collection.**
(DOCX)

**S1 Checklist. Inclusivity in global research.**
(DOCX)

## Acknowledgments

This manuscript is the result of a study conducted in the framework of the International PhD in Global Health, Humanitarian Aid, and Disaster Medicine organized by Università del Piemonte Orientale (UPO).

We would like to express our sincere gratitude to all our study participants, without whom this research would not have been possible. We are deeply thankful to the cultural mediators of *Farsi Prossimo* for their tireless support throughout the study. We also extend our appreciation to the healthcare workers at Villa Marelli and the Italian Red Cross, whose dedicated efforts are essential to the TB screening programs in our region. A special thanks to the personnel of Villa Marelli TB Reference Center for their continued support, and to Martina Valente for her kind guidance and valuable contributions to this work.

Copan Italia S.p.A (Brescia, Italy) provided FLOQSwabs free of charge. Copan had no role in study design, collection, analysis, or interpretation of data.

## Author contributions

**Conceptualization:** Renée Codsi, Francesca Saluzzo, Gerard A Cangelosi, Daniela Maria Cirillo.

**Data curation:** Renée Codsi, Francesca Saluzzo.

**Formal analysis:** Renée Codsi, Francesca Saluzzo.

**Funding acquisition:** Gerard A Cangelosi, Daniela Maria Cirillo.

**Investigation:** Renée Codsi, Francesca Saluzzo.

**Methodology:** Renée Codsi, Francesca Saluzzo, Giulia Russo, Ramya Kumar, George Wanje.

**Project administration:** Francesca Saluzzo, Giulia Russo, Luigi R Codecasa, Giovanni Fumagalli, Daniela Maria Cirillo.

**Supervision:** Luca Ragazzoni, Gerard A Cangelosi, Daniela Maria Cirillo.

**Visualization:** Francesca Saluzzo.

**Writing – original draft:** Renée Codsi, Francesca Saluzzo.

**Writing – review & editing:** Renée Codsi, Francesca Saluzzo, Rachel C Wood, Alaina M. Olson, George Wanje, Marlana Kohn, Luigi R Codecasa, Giovanni Fumagalli, Daniela Maria Cirillo.

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
