## [Decision Letter · Decision Letter 0]

30 Sep 2025

PGPH-D-25-01621

Acceptability of Tongue Swabs for Tuberculosis Screening in Migrant Settings in Northern Italy: A Qualitative Study

Dear Dr. Cirillo,

Thank you for submitting your manuscript to PLOS Global Public Health. After careful consideration, we feel that it has merit but does not fully meet PLOS Global Public Health’s publication criteria as it currently stands. Therefore, we invite you to submit a revised version of the manuscript that addresses the minor points raised during the review process.

We look forward to receiving your revised manuscript.

Kind regards,

Miguel Angel Garcia-Bereguiain, PhD

Academic Editor

Journal Requirements:

2. Please provide a detailed online Financial Disclosure statement. This is published with the article. It must therefore be completed in full sentences and contain the exact wording you wish to be published.

a) State the initials, alongside each funding source, of each author to receive each grant, if applicable. For example: “This work was supported by the National Institutes of Health (####### to AM; ###### to CJ) and the National Science Foundation (###### to AM).”

For more information, please see our guidelines: https://journals.plos.org/globalpublichealth/s/submission-guidelines#loc-financial-disclosure-statement

3. Please ensure that the funders and grant numbers match between the Financial Disclosure field and the Funding Information tab in your submission form. Note that the funders must be provided in the same order in both places as well.

4. Please update your online Competing Interests statement. If you have no competing interests to declare, please state: “The authors have declared that no competing interests exist.”

5. In the online submission form, you indicated that “The data supporting the findings of this study are not publicly available but can be obtained from the corresponding author upon reasonable request.”.

a) In a public repository,

b) Within the manuscript itself, or

c) Uploaded as supplementary information.

6. We have noticed that you have uploaded Supporting Information files, but you have not included a list of legends. Please add a full list of legends for your Supporting Information files before or after the references list.

7. Some material included in your submission may be copyrighted. According to PLOS’s copyright policy, authors who use figures or other material (e.g., graphics, clipart, maps) from another author or copyright holder must demonstrate or obtain permission to publish this material under the Creative Commons Attribution 4.0 International (CC BY 4.0) License used by PLOS journals. Please closely review the details of PLOS’s copyright requirements here: PLOS Licenses and Copyright. If you need to request permissions from a copyright holder, you may use PLOS's Copyright Content Permission form.

Potential Copyright Issues:

Images in “Semi structured interview guide_ migrants .docx”: Please confirm whether you drew the images / clip-art within the figure panels by hand. If you did not draw the images, please provide (a) a link to the source of the images or icons and their license / terms of use; or (b) written permission from the copyright holder to publish the images or icons under our CC-BY 4.0 license. Alternatively, you may replace the images with open source alternatives. See these open source resources you may use to replace images / clip-art:

- https://openclipart.org/

8. We noticed that you used “Unpublished master’s thesis” in the manuscript. We do not allow these references, as the PLOS data access policy requires that all data be either published with the manuscript or made available in a publicly accessible database. Please amend the supplementary material to include the referenced data or remove the references.

9. We do not publish any copyright or trademark symbols that usually accompany proprietary names, eg (R), (C), or TM (e.g. next to drug or reagent names). Please remove all instances of trademark/copyright symbols throughout the text, including ® on page 20.

Additional Editor Comments (if provided):

Reviewers' comments:

Reviewer's Responses to Questions

**Comments to the Author**

1. Does this manuscript meet PLOS Global Public Health’s publication criteria?

Reviewer #1: Yes

Reviewer #2: Yes

2. Has the statistical analysis been performed appropriately and rigorously?

Reviewer #1: N/A

Reviewer #2: N/A

3. Have the authors made all data underlying the findings in their manuscript fully available (please refer to the Data Availability Statement at the start of the manuscript PDF file)?

Reviewer #1: No

Reviewer #2: No

4. Is the manuscript presented in an intelligible fashion and written in standard English?

Reviewer #1: Yes

Reviewer #2: Yes

Reviewer #1: I believe this paper represents a valuable contribution to the literature on TB diagnosis and migrant health from a qualitative perspective. Overall, it is well written and substantiated, highlighting supervised self-swabbing (SSS) and the use of the COM-B model as strengths. This approach could even be applied to other respiratory infections where current sample collection methods are invasive or impractical.

I have the following minor comments:

- Only 24 interviews were conducted, with a strong gender imbalance (19 men, 5 women). This limits generalizability, particularly given potential gendered differences in health-seeking behavior. Please mention this in the discussion paragraph where the limitations of the study are discussed.

- Recruitment challenges with migrants from West Asia and North Africa are acknowledged but deserve further elaboration. How might this underrepresentation bias results?

- While rapid qualitative analysis is justified, a deeper description of how disagreements were resolved during coding would strengthen transparency.

- The discussion sometimes overstates the potential of SSS without equally weighing concerns about its diagnostic sensitivity compared to sputum. Although the focus is on acceptability, readers may expect a more balanced perspective on clinical utility.

- It would be helpful to more explicitly address whether findings are generalizable beyond Northern Italy, especially to higher TB burden or lower-resource migrant-receiving settings.

- Improve graphs quality (figures 1 and 2). These should not be pixelated.

Reviewer #2: The topic is highly relevant and innovative, as it explores tongue swabs as a practical alternative to sputum collection, particularly in migrant settings.

Minior Concerns.

1. It is important to mention that, as the study was conducted in a single referral center, the findings may not be fully generalizable to other migration contexts in Italy or Europe.

2. The conclusion could be strengthened with a clearer call for public policies that support the integration of multilingual, culturally adapted audiovisual materials into TB screening programs.

**Do you want your identity to be public for this peer review?** For information about this choice, including consent withdrawal, please see our Privacy Policy

Reviewer #1: **Yes:** Ángel Sebastián Rodríguez-Pazmiño

Reviewer #2: **Yes:** Elsy Carvajal

---

## [Editor Report · Decision Letter 1]

16 Dec 2025

Acceptability of Tongue Swabs for Tuberculosis Screening in Migrant Settings in Northern Italy: A Qualitative Study

PGPH-D-25-01621R1

Dear Dr. Cirillo,

We are pleased to inform you that your manuscript 'Acceptability of Tongue Swabs for Tuberculosis Screening in Migrant Settings in Northern Italy: A Qualitative Study' has been provisionally accepted for publication in PLOS Global Public Health.

Best regards,

Miguel Angel Garcia-Bereguiain, PhD

Academic Editor